# Potential Application of Graphene/Antimonene Herterostructure as an Anode for Li-Ion Batteries: A First-Principles Study

**DOI:** 10.3390/nano9101430

**Published:** 2019-10-10

**Authors:** Ping Wu, Peng Li, Min Huang

**Affiliations:** 1School of Physics and Engineering, Zhengzhou University, Zhengzhou 450001, China; wup@mail.ustc.edu.cn; 2School of Electrical and Electronic Information, Shangqiu Normal University, Shangqiu 476000, China; plisurfeng@126.com; 3Key Laboratory of Ferro and Piezoelectric Materials and Devices of Hubei Province, Faculty of Physics and Electronic Sciences, Hubei University, Wuhan 430062, China

**Keywords:** graphene/antimonene heterostructure (G/Sb), Li adsorption properties, diffusion energy barrier, theoretical specific capacity, first-principles calculations

## Abstract

To suppress the volume expansion and thus improve the performance of antimonene as a promising anode for lithium-ion batteries, we have systematically studied the stability, structural and electronic properties of the antimonene capped with graphene (G/Sb heterostructure) upon the intercalation and diffusion of Li atoms by first-principles calculations based on van der Waals (vdW) corrected density functional theory. G/Sb exhibits higher Young’s modulus (armchair: 145.20, zigzag: 144.36 N m^−1^) and improved electrical conductivity (bandgap of 0.03 eV) compared with those of antimonene. Li favors incorporating into the interlayer region of G/Sb rather than the outside surfaces of graphene and antimonene of G/Sb heterostructure, which is caused by the synergistic effect. The in-plane lattice constants of G/Sb heterostructure expand only around 4.5%, and the interlayer distance of G/Sb increases slightly (0.22 Å) at the case of fully lithiation, which indicates that the capping of graphene on antimonene can effectively suppress the volumetric expansion during the charging process. Additionally, the hybrid G/Sb heterostructure has little influence on the migration behaviors of Li on the outside of graphene and Sb surfaces compared with their free-standing monolayers. However, the migration energy barrier for Li diffusion in the interlayer region (about 0.59 eV) is significantly affected by the geometry structure, which can be reduced to 0.34 eV simply by increasing the interlayer distance. The higher theoretical specific capacity (369.03 mAh g^−1^ vs 208 mAh g^−1^ for antimonene monolayer) and suitable open circuit voltage (from 0.11 V to 0.89 V) of G/Sb heterostructure are beneficial for anode materials of lithium-ion batteries. The above results reveal that G/Sb heterostructure may be an ideal candidate of anode for high recycling–rate and portable lithium-ion batteries.

## 1. Introduction

Owing to their high reversible capacity, high energy/power density, and long lifespan, lithium-ion batteries (LIBs) have received considerable interests for electrochemical energy storage and conversion devices such as electric vehicles and portable electronic devices [1,2,3]. In the past decade, two-dimensional (2D) materials, such as graphene [4], MoS_2_ [5], black or blue phosphorene [6,7], SnP_3_ [8], and MXenes [9], have attracted significant attention as promising anode material in metal ion batteries, which exhibit enhanced electrochemical properties compared to their bulk counterparts due to their large surface-to-volume ratio, fast ion diffusion, and enlarged surface active sites for ion storage. However, the low adsorption energy and the weak binding strength between Li and 2D layered materials significantly hampered their practical applications in LIBs.

Recently, antimonene (Sb) possessing a buckled hexagonal structure has been successfully fabricated by mechanical exfoliation [10], van der Waals epitaxy on the PdTe_2_ [11], as well as liquid-phase exfoliation [12]. Due to its low price, high theoretical capacity, moderate working voltage, and unique puckered-layer structure, antimonene has become an attractive choice as an anode material for high-performance LIBs and sodium-ion batteries (SIBs) [13,14,15,16]. The capacity of a few layers of antimonene for Li storage can be maintained at 584.1 mAhg^−1^ after 100 charging/discharging cycles, and the capacity fading during 100 cycles was only 3.8%, which is superior to those of Sb bulk and Sb nanoparticles [15]. It is also reported by Su et al. that ML antimonene favors a low Li diffusion energy barrier of 0.34 eV along the in-plane direction, which is also a critical indicator to obtain high-performance LIBs and supercapacitors based on the antimonene-electrode [14]. However, the utilization of Sb anode is seriously restricted by the severe volumetric change during the chargeable/dischargeable processes, such as the considerable volume expansion (~200% for LIBs) caused by Li intercalation [16], which may lead to severe mechanical pulverization of anode materials, rapid capacity fading, and the cycling lifetime shortening of rechargeable batteries. Therefore, it is anticipated to design novel anode materials based on antimonene to suppress the volumetric changes.

The vertical heterostructure stacked by different 2D materials is a promising way to construct new materials [17] that integrates the properties of their isolated components, which is expected to be beneficial for rapid electron transport as well as accelerated cation transport in electrodes and further improve the rate performance of LIBs and SIBs. Due to its fascinating electrical conductivity and excellent mechanical properties, graphene is commonly introduced to construct hybrid electrode materials, which exhibits attractive electrochemical properties and high structural stability [18,19,20,21,22,23,24,25,26,27,28]. A series of experimental and theoretical study have confirmed that graphene hybrid with MoS_2_ [18], black phosphorene (black-P) [19,20], and transition metal carbides (MXenes) [21], as well as metal oxides, have shown good performance in LIBs. Furthermore, based on the high specific surface area and flexibility of graphene, good interfacial contact between particles and capping graphene could effectively prevent the Li aggregation during volume contraction/expansion and reduce the volume changes caused by an alloying reaction [22]. For example, although blue phosphorene (blue-P) tended to be oxidized under ambient condition, the capping layer of graphene on blue-P can effectively protect blue-P from oxidation, which has been demonstrated theoretically to be good anode materials for LIBs [23]. An obvious advantage for a vertical heterostructure is that the interlayer distance is feasible and can be adjusted to intercalate as many metal ions as possible and further improve the specific capacity of batteries [24,25]. A high specific capacity of more than 5000 mAh g^−1^ [24] has already been demonstrated for batteries with unstacked carbon nanosheets implanted by an N-doped carbon nanotube (CNT) forest as electrodes, which indicated that the pillared layered structures with larger interlayer distance could effectively improve the capacity. Moreover, compared to the semiconducting black-P, MXenes, and TMSs such as MoS_2_ and WS_2_, the introduction of graphene could effectively enhance the electron transport properties of hybrid composites as anode materials, which is an important influence factor for high-performance LIBs. For instance, hybrid black-P/graphene (P/G) exhibits a small bandgap of 9 meV, which mainly arise from bandgap opening of graphene owing to the breaking of structural symmetry caused by the black-phosphorene interaction [20]. It has also been reported that the adsorption energy of Li atoms and mechanical stiffness of phosphorene/graphene or MXene/graphene can be effectively enhanced [20,26,27], and the cycling stability of MoS_2_ can be improved [28] by capping of graphene on phosphene, MXene and MoS_2_, respectively.

Inspired by the graphene/blue-P, graphene/MXenes, graphene/MoS_2_ pillared structure of Ti_3_C_2_O_2_ intercalated by quinine molecules [25], we introduced the chemically stable graphene as an effective capping agent for antimonene to design a unique hybrid graphene/antimonene heterostructure (G/Sb) as promising anode materials for LIBs. Zhang et al. studied the electronic properties of G/Sb heterostructure by DFT calculations and predicted that the band structure of the G/Sb heterostructure seems to be a simple sum of those of graphene and antimonene showing a zero band gap [29]. Meanwhile, G/Sb heterostructure has been successfully synthesized and applied in anode of sodium-ion batteries, which showed high Na capacity and excellent cycling performance [30]. Therefore, it is expected that the graphene can greatly improve the electrical conductivity of the hybrid structure and effectively prevent the large volume expansion of antimonene caused by the allying reaction. In this paper, we systematically investigated Li intercalation in G/Sb heterostructure for rechargeable battery applications. We focus on the stability of G/Sb and the Li adsorption and diffusion properties on the G/Sb performed by first-principles calculations based on vdW corrected density functional theory. Furthermore, we also estimate the specific capacity and average open circuit voltage of the G/Sb heterostructure. 

## 2. Methods

All calculations were performed based on the densityfunctional theory (DFT) as implemented in the Vienna ab initio simulation package (VASP) [31,32]. The exchange correlation energy was described by the generalized gradient approximation (GGA) in the scheme proposed by the Perdew-Burke-Ernzerhof (PBE) functional [33]. The projector augmented wave (PAW) method is employed to describe electron–ion interactions [34]. The plane wave basis set with cutoff energy of 450 eV is used. The vacuum layer of 20 Å was set to eliminate the interactions between adjacent supercells. Interlayer vdW interactions of the graphene/antimonene (G/Sb) systems are considered in our calculations by using a dispersion correction term with the DFT-D_2_ method of Grimme [35]. The k-points grids of 5 × 5 × 1 and 9 × 9 × 1 with Γ-center schemes [36] were used to sample the first Brillouin zone for structural optimizations and electronic calculations, respectively. After full optimization, the lattice constants of graphene and antimonene are 2.47 and 4.12 Å, respectively, which are consistent with previous studies [4]. To reduce the lattice mismatch between the layers, the G/Sb heterostructure are constructed by a supercell with (5 × 5) graphene and (3 × 3) antimonene, which includes 50 carbon atoms and 18 antimony atoms. Several possible configurations with different relative positions of graphene and antimonene shown in Figure 1 are considered in the search for the most stable configure of G/Sb heterostructure (with lowest total energy and formation energy).

To investigate the diffusion properties of Li in the G/Sb heterostructure, the nudged elastic band (NEB) method was adopted to calculate the transition state and migration barrier [37,38]. All the geometry structures are fully relaxed until the total energies satisfied the convergence of 1.0 × 10^−5^ eV and forces on each atom smaller than 0.01 eV/Å, respectively.

## 3. Results and Discussion

### 3.1. Stability, Structural, and Electronic Properties of G/Sb Heterostructure

The geometrical structures of monolayer graphene (G) and antimonene (Sb) are optimized for references. Graphene exhibits hexagonal honeycomb structure with lattice constant of 2.47 Å and its C–C bond length and bond angle are 1.43 Å and 120°, respectively. However, in order to increase the stability of layered structure, antimonene adopts buckled honeycomb structures with lattice parameter and layer thickness of 4.12 Å and 1.65 Å, which is similar to those of buckled arsenene and silicene [39,40]. The above calculated results are in good agreement with previous results.

In order to find out the most stable stacking pattern for the graphene on antimonene, here, we considered four typical stacking configurations, H-H, H-C, Sb-C, and Sb-H_G_, for G/Sb heterostructure as illustrated in Figure 1. All systems were geometrically optimized to get the most stable atomic configuration. We introduced the formation energy to investigate the thermodynamic stability of possible stacking configuration for G/Sb heterostructure. The formation energy per atom was described as follows:(1)$$E_{f}=(E_{G/Sb}^{total}-E_{G}^{total}-E_{Sb}^{total})/N,$$
where $E_{G/Sb}^{total}$ is the total energy of G/Sb heterostructure and $E_{G}^{total}$ and $E_{Sb}^{total}$ represent the total energies of the pristine graphene and antimonene monolayers, respectively. *N* is the total number of atoms in the system. The results for bilayer graphene and bilayer antimonene are also calculated as references (listed in Table 1). Our calculated interlayer formation energy of bilayer graphene with AB stacking pattern are −24.95 meV per atom, which is in good agreement with previous studies of −18 meV per atom obtained by quantum Monte Carlo simulations [21]. AB stacking is the ground state structure for bilayer antimonene, and *E_f_* is −65.12 meV per atom due to the relatively less Sb atoms per unit area. After fully relaxation, the formation energy for the H–H stacking configuration of G/Sb heterostructure (shown in Figure 2a) with lowest total energy among four geometries considered is about −30.43 meV per atom, implying that the G/Sb heterostructrure exhibits high thermodynamic stability similar to that of bilayer graphene. Besides, the Sb–Sb bond length (2.89 Å) changes little while the C–C bond length (1.43 Å) is slightly shorter than that in freestanding graphene (1.46 Å). The equilibrium interlayer distance (3.53 Å) is slightly shorter than those of P/G (3.61 Å) [20], C_3_N/P (3.61 Å) [41], while larger than that of bilayer antimonene (2.84 Å) and bilayer graphene (3.28 Å) with AB stacking pattern in our work. Additionally, deformation resistance is closely associated with the cycling performance of batteries, so we also investigated the in-plane mechanical properties of the G/Sb heterostructure. The calculated results show that the Young’s modulus of the G/Sb heterostructure (armchair: 145.20, zigzag: 144.36 N m^−1^) are greatly enhanced compared with those of monolayer antimonene (armchair: 16.70, zigzag: 13.29 N m^−1^), which can be explained by the synergy effect between individual monolayers. This improved stiffness of G/Sb is rather vital as anode materials for LIBs due to the large stresses and deformations resistance caused by the lithium intercalation.

The band structures and density of states (PDOS) of monolayer graphene, antimonene, and G/Sb are shown in Figure 2c,d. The pristine antimonene presents a semiconductor character with a direct band gap of 1.26 eV, which agrees well with previous result (1.04 eV) of antimonene [42]. When antimonene is capped by graphene, the G/Sb heterostructure with much smaller bandgap of 0.03 eV exhibits electrical conductivity superior to that of antimonene, in which graphene can provide an electrical conducting channel even at a low lithiation level. Previous studies have confirmed that anode materials with relative small band gaps such as graphene/black-phosphorene and graphene/MoS_2_ or metallic characters (e.g. blue-P/MS_2_(M = Nb,Ta) heterostructure) [43] are of great benefit for the conductivity of electrode materials. Hence, G/Sb with good electrical conductivity may become a viable candidate for anode materials of LIBs.

To further give insight into how the coating of graphene affects the electronic properties of the G/Sb system, we also investigated the charge transfer and charge density difference distributions. The planar averaged charge density difference along the z direction is plotted in Figure 2c, which is defined as
(2)$$\Delta \rho_{z}=\rho_{z}(G/Sb)-\rho_{z}(G)-\rho_{z}(Sb),$$
where $\rho_{z}(G/Sb)$ is the planar averaged density along z direction of G/Sb, $\rho_{z}(G)$ and $\rho_{z}(Sb)$ are those of the pristine graphene and antimonene with the same positions as in the heterostructure. This charge distribution shows that there are two depletion regions around the topmost graphene surface atoms and a significant accumulation region near the Sb atoms located the upper positions, indicating that the remarkable transfer of charge mainly exists in the interface between graphene and antimonene. In an effort to quantitatively estimate the amount of charge transfer between graphene and antimonene in G/Sb, Bader charge analysis [44,45] was carried out in our study. The calculated results show that the charge transfer of 0.139|e| occurs from the antimonene surface to the top-most C atoms (as shown in insert of Figure 2b), which can be understood by the fact that the C atoms are more electronegative than the Sb atoms. 

### 3.2. Adsorption and Intercalation of Single Li in the G/Sb Heterostructure

To examine the properties of Li incorporating into the G/Sb heterostructure, several possible adsorption and intercalation sites were considered in our study. We have explored three different regions—(1) Li adsorbed on the outside graphene surface (Li/G/Sb), (2) Li adsorbed on the outside antimonene surface (G/Sb/Li), and (3) Li intercalated in the interlayer of graphene and antimonene of G/Sb heterostructure (G/Li/Sb). Nine adsorption and intercalation sites of Li adatom are considered to search for the most stable geometries. For the cases of Li adsorbed on the outside graphene surface (Li/G/Sb), two adsorption sites considered are *H*_C_ (above the center of hexagonal C-C rings) and *T*_C_ (the top site of C atom), as shown in Figure 3a, while for the case of G/Sb/Li, we considered three high-symmetrical adsorption sites including *T*_Sb_(the top site of lower Sb atom), *T*^’^_Sb_(the top site of upper Sb atom), and *H*_Sb_(above the center of hexagonal Sb-Sb rings), as observed in Figure 3b. For the intercalation of Li in the interlayer of G/Sb heterostructure, there were four possible adsorption sites—the center of the hexagonal C-C rings which is near the *H*_Sb_ site (*I*_H1_), the two-fold site of *H*_C_ and *H*_Sb_ (*I*_H2_), the two-fold site of *T*_C_ and *H*_Sb_ (*I*_c_), and the top site of lower Sb atom, which is near the *H*_C_(*I*_Sb_), as illustrated in Figure 3c. For comparison, adsorption of Li on the freestanding graphene and antimonene are also considered. 

To better compare the adsorption behaviors of Li at different adsorption and intercalation sites, we calculate the binding energy, which is defined as
(3)$$E_{b}=(E_{G/Sb+Li}-E_{G/Sb}-nE_{Li})/n,$$
where $E_{G/Sb+Li}$ and $E_{G/Sb}$ denote the total energy of the G/Sb heterostructures with and without Li adsorbed or intercalated, respectively, and $E_{Li}$ is the energy of an isolated Li atoms, *n* is the number of Li atoms adsorbed or intercalated in the G/Sb heterostructures. According to the definition, a larger value of *E*_b_ represents a stronger bonding of Li to the G/Sb heterostructures. The calculated results for the corresponding structural and energetic parameters are summarized in Table 2. Our results can be summarized as follows. First, the *H*_C_ and *T*_Sb_ are the most favorable adsorption sites for Li/G/Sb and G/Sb/Li system, respectively, which are identical to the adsorption cases of Li adatom on the pristine monolayers (Li/G and Li/Sb). However, the adsorption behavior of Li adatoms on the antimonene surface can be remarkably enhanced by increasing the binding energy from 1.96 eV for pristine antimonene to 2.23 eV for Li adsorption at *T*_Sb_ sites, while the change of *E_b_* for Li/G/Sb is only 0.10 eV compared with the case of Li/G. Second, the binding energies for Li occupying different sites of the interlayer of G/Sb are in the order of *I*_C_(2.67 eV) < *I*_H1_(2.80 eV) < *I*_H2_(2.88 eV) < *I*_Sb_(2.92 eV). It is obvious that *I_Sb_* is the most stable site for Li intercalated in the interlayer site of G/Sb, since it is very close to the most stable sites of Li adsorbed on the outside graphene surface (*H_c_*) and the antimonene surface (*T_Sb_*), indicating that the stability of the intercalation of Li in G/Sb depends on both graphene and antimonene. Finally, it can be seen that the G/Li/Sb system is energetically more stable than Li/G/Sb and G/Sb/Li systems by 0.89 eV and 0.69 eV, respectively, which is ascribed to the interaction of Li atom with both graphene and antimonene at the interlayer sites. These calculated results indicate that the Li adatom prefers to reside in the interlayer space of G/Sb heterostructure rather than be incorporate at the outside surfaces of graphene and antimonene during the lithiation process. In other words, Li will first intercalate into the interlayer of the G/Sb during the lithiation process and then occupy the outside surfaces of graphene and antimonene in G/Sb, which is similar to the case of G/black-P [23] but is different from the case of blue-P/MS_2_ (M = Nb,Ta) heterostructure [43]. Furthermore, the significant enhancement of Li bonding strength in the G/Sb systems compared to the pristine graphene and antimonene ML is mainly contributed by the synergistic effects between graphene and antimonene, which is beneficial to improve the Li capacities. The enhanced binding energy between Li and G/Sb not only avoid the formation of metallic lithium but also increases the security and reversibility of LIBs, which make it possible for G/Sb to be the candidate anode materials for LIBs in experiments.

For the case of G/Li/Sb and G/Sb/Li, a significant charge transfer from Li atom to the neighboring C and Sb atoms, which is due to the atomic electronegativity in the order of C > Sb > Li. While the Li atom adsorbed at the outside surface of graphene, the C atoms are accepters gaining about 1|e| from both Li and Sb, which verify the strong ionic interaction of Li with graphene and antimonene. Furthermore, the large amount of electron transfer listed in Table 2 indicates that the Li atoms are strongly polarized after adsorption, and Coulomb force dominates the interactions between Li and G/Sb. Additionally, the charge transfer (Δ*Q*) for G/Li/Sb and G/Sb/Li are similar to those on ML antimonene (Li/Sb), which indicates that graphene as capping layer does not influence the electronic properties of antimonene. 

### 3.3. Effects of Li Concentration on the Adsorption Stability and Volume Expansion of G/Sb Heterostructure

It is known that the performanceof LIBs relies greatly on the Li capacity of the anode materials. Therefore, it is of great importance to understand the electrochemical properties of different lithiated phases during the Li intercalation process by adding more Li atoms to the G/Sb heterostructure. The most stable structures of Li*_x_*C_50_Sb_18_ (*x* = 1, 3, 6, 9, 18, and 28) in the process of Li intercalation were explored. For each concentration, we calculated several different configurations. After full geometry optimization, the variation of the averaged binding energy (*E_b_*) as a function of the number of Li ions for the corresponding most stable configurations are shown in Figure 4. As the additional Li atoms are further inserted into the G/Sb, the average *E_b_* decreases gradually from 2.73 eV to 2.43 eV with the number of inserted Li ions increasing from three to 28, which indicated that the G/Li/Sb structures are stable even at high Li concentrations. We found that the charge transfer from the Li atoms to the G/Sb at higher concentrations was reduced compared to that of lower concentrations. When the number of embedded Li increases, the interatomic distances between positively charged Li ions is also reduced. The reduction trend on the binding energy can be attributed to two main factors—(1) the weak electrostatic attraction between the G/Sb heterostructure and the Li cations, which is closely related to the reduction of charge transfer from Li to G/Sb at high concentrations, and (2) the enhanced Li–Li repulsion at high Li concentrations, which is resulted from the shorter Li–Li distance.

It is reported that the performance of Sb anode is seriously restricted by its volume expansion due to Li intercalation [16], so we studied the structural variation of G/Li/Sb with the increasing of the Li ion concentration intercalated. Compared with the equilibrant interlayer distance (3.53 Å) of clean G/Sb, as listed in Table 2, the distances between graphene and antimonene (*d* = 3.44–3.45Å) was slightly reduced when an isolated Li adsorbed on outside antimonene surfaces of G/Sb, while the *d* values (3.50–3.54 Å) change little except for *I_Sb_* configuration with Li intercalated into the interlayer of G/Sb. Taking the case of fully lithiation as an example, we found that the in-plane lattice constant of G/Li/Sb systems present a slight lattice expansion of around 4.5% along the *a/b*-axis direction as the number of Li ions is increased from 0 to 28, which is significantly smaller than the 1.22-fold expansion in-plane channel reported by experiments for few layer antimonene [15]. Besides, we did not find significant distortion for G/Li/Sb heterostructure even when the number of Li reached to 28. We also did not observe any severe change of the C–C and Sb–Sb bonds of G/Li/Sb heterostructure with the increasing of Li concentrations. It was also found that the interlayer distance between graphene and antimonene slightly increases at low Li concentrations (*x* ≤ 3) and then gradually increases relatively significantly when additional Li atoms constantly insert into the interlayer (3< *x* ≤ 9). This fact can be explained by the new interaction mechanism of C–Li–Sb at the low concentration instead of the weak vdW interaction in G/Sb heterostructure, but the enhanced Li-Li repulsion and increasing number of the Li adatoms will lead to the separation of two monolayers at a high concertation. The calculated results on interlayer separation suggest that Li intercalation could cause a slight expansion of 0.22 Å in the vertical direction of the G/Sb heterostructure, which is quite a bit smaller than expansion of few layer antimonene along vertical direction (1.14 Å) during the sodiation process reported in experiments [13]. Therefore, the presence of coating graphene with unique mechanical properties on antimonene can effectively prevent the volume expansion of antimonene. These results clearly illustrated that the layered G/Sb heterostructure possess a reversible reaction process during intercalation and removing of Li, which is beneficial for rechargeable ion batteries. These findings provide an effective method to overcome the non-circularity performance and volume expansion problem faced by antimonene as electrode materials in experiments. It is clearly demonstrated that G/Sb can be utilized as a new kind of anode material to improve the performance of antimonene for high-capacity LIBs.

### 3.4. Energy Performance of G/Sb Heterostructure

To get insight into the interaction between intercalated Li and the G/Sb heterostructure, we investigate the partial density of state (PDOS) of G/Sb heterostructure with 1, 9, 18, and 28 Li incorporated as depicted in Figure 5. Upon the Li intercalation into G/Sb, the system exhibits metallic character, which is different from the semiconducting behavior of ML antimonene, indicating the G/Sb obtained electron from foreign Li atoms. Similar phenomenon has also been observed in previous studies on the graphene/MoS_2_ and graphene/black-P heterostructures [19,20]. It should be noted that the transition from semiconducting antimonene to metallic G/Sb has significant importance for its application in LIBs. In all cases of different Li concentrations, it is obvious that the orbitals of Li overlap with the orbitals of both graphene and antimonene, which indicated that the intercalated Li simultaneously formed strong covalent hybridization with graphene and antimonene. Although the bond between Li and Sb atoms is dominated by ionic interactions, some covalent components still make a minor contribution. 

Open circuit voltage (OCV) and specific capacity are commonly used to estimate the performance of batteries. Under the premise that the small change in volume and entropy are ignored, the OCV can be approximately computed from the total energy difference based on the following equation:(4)$$OCV\approx \frac{E_{G/Sb+x_{1}Li}+(x_{2}-x_{1})\mu_{Li}-E_{G/Sb+x_{2}Li}}{(x_{2}-x_{1})e},$$
where $E_{G/Sb+x_{1}Li}$ and $E_{G/Sb+x_{2}Li}$ denote the total energies of supercell of the G/Sb with *x*_1_ and *x*_2_ Li atoms intercalated, respectively, and $\mu_{Li}$ is the chemical potential of Li atom, which is equal to the energy of a Li atom in its bulk bcc crystal structure. *e* is the unit of charge. The calculated OCV of G/Sb as a function of the Li concentrations and the corresponding optimized configurations are shown in Figure 6. It can be seen that the OCV decreases from 0.89 V to 0.11 V as the Li concentrations increases during the process of lithiation, which is within the desired potential range (0.10–1.00 V) for anode materials [20]. It is known that the relatively low OCV is necessary for G/Sb as anode for LIBs. When the value of OCV reaches its minimal, the concentration of Li incorporated in G/Sb heterostructure reaches its peak value, which corresponds to the case of full lithiation.

Li storage capacity is another factor to determine the performance of LIBs, and the theoretical specific capacity can be given by
(5)$$C\approx \frac{xF}{M_{G/Sb}},$$
where *x* is the maximum number of Li adatoms intercalated, F is the Faraday constant (26.081 Ah mol^−1^), and $M_{G/Sb}$ is the mole weight of the G/Sb heterostructure. It is expected that the G/Sb heterostructure can offer larger storage capacity than bilayer antimonene since the G/Sb heterostructure has a smaller mole weight compared to bilayer antimonene. According to Figure 5, we can estimate the maximum number of Li adatoms intercalated to G/Sb is 28. Based on the Equation (5), the theoretical specific capacity of G/Sb are estimated to be 270.13 mAh g^−1^, which is higher than that (208 mAh g^−1^) of pristine antimonene in reference [14]. The above results show that the moderate open circuit voltage and high specific capacity of G/Sb make it a potential anode material for LIBs.

The theoretical specific capacity of G/Sb obtained is smaller than the experimental values for a few layers of antimonene and Sb nanoparticles, but antimonene rapidly expands about 200% in volume [15]. With the capping of graphene, our calculations predicted that the volume expand is slight (around 4.5% at in-plane direction and increasing of 0.22 Å for interlayer distance) in the case of full lithiation, which is in agreement with G/Sb heterostructure as anode of sodium-ion batteries [30]. The theoretical specific capacity of G/Sb can be increased to be 549.91 mAh g^−1^, which is close to the experimental value for few layer antimonene [15] by intercalating more Li ions in G/Sb heterostructures in consequence of notorious volume changes.

### 3.5. Li Diffusion on the G/Sb Heterostructure

The rate capacity is the essential issue for high-performance LIBs, which is closely associated with the high electrical conductivity and Li mobility. It has been discussed that the presence of graphene could significantly reduce the bandgap of antimonene, and the incorporation of Li atoms further improves the electrical conductivity of G/Sb. Since the fast Li diffusion is vital for the battery to rapidly charge/discharge, we next turn to investigate the mobility of Li atoms at G/Sb heterostructure. As listed in Table 2, the interactions between Li and G/Sb heterostructure are much stronger than those between Li and graphene or antimonene, and the adsorption sites and diffusion pathways in the G/Sb systems thus mainly occur in the following regions: on the outside surface of graphene or antimonene, from the outside surface of graphene or antimonene to interlayer space, and the interlayer space between graphene and ML antimonene. The optimized migration pathways and corresponding energy profiles are presented in Figure 7.

On the outside surfaces of graphene, it is noted that the energy barrier is about 0.35 eV for diffusion of Li adatom from *H_C_* site to the nearest-neighboring *H_C_* site in Figure 7a, which is comparable to the cases of Li diffusion on pristine graphene in previous studies (0.32 eV) [4]. Similarly, we also found that the Li adatom shared the same migration pathway both on outside surface of antimonene of G/Sb heterostructure and ML antimonene, such as diffusing from *T*_Sb_ site to the nearby *T*_Sb_ site passing through the intermediate stable *H*_Sb_ site as shown in Figure 7b. The calculated diffusion barrier is slightly higher by 0.03 eV than that for the diffusion of Li (0.34 eV) on ML antimonene reported by Sengupta et al. [14]. By comparison, we found that the diffusion behavior of Li adatom on the outside surface of graphene and antimonene is almost unaffected by the bilayer heterostructure, which is stacked by weak vdW interaction.

As listed in Table 2, Li adatom inserted into the interlayer of G/Sb is more stable than that adsorbed at the outside surface of graphene or antimonene. Therefore, we also investigated the possibility of Li migration from the outside surface to the interlayer region by directly crossing the graphene and antimonene surfaces, which are shown in Figure 7c,d. It is noticeable that the energy barrier is about 10.51 eV for Li diffusing through the C–C ring, which is consistent with the value (10.02 eV) of Li adatom crossing the graphene in previous study [46]. The diffusion barrier for Li diffused from *H*_Sb_ site to *I*_C_ site through the Sb–Sb ring is about 1.26 eV, which is compatible to the case of Li migrate cross Sb layers in few layer antimonene (1.14 eV) [15] and in the bulk Sb (1.73 eV) [47]. These calculated results indicate that it is much easier for Li passing through antimonene to accommodate at the most stable sites at the interlayer region of G/Sb than through graphene. We think the different results are mainly caused by Sb–Sb ring with larger size of about 4.76 Å (2.85 Å for C–C ring), which is conducive to the passing of Li via the center of the hexagon. Interestingly, Kistanov and co-workers found that the tensile strain could effectively reduce the energy barrier for a lithium atom to diffuse across the antimonene sheet based on the DFT calculations [48], which can further confirm our calculations. 

For the cases of Li adatom diffusion in the interlayer space of G/Sb, two possible adsorption stable sites (*I_Sb_* and *I_H2_*) were considered. The diffusion of the Li adatoms is mainly through two possible routes, namely path-I and path-II shown in Figure 7e. For path-I (left panel of Figure 7e), the Li atom diffuses between two adjacent *I_Sb_* sites across the neighboring *I_C_* site as an intermediate state, which is quite similar to the case of the pristine ML antimonene. For path-II (right panel of Figure 7e), Li atoms move from site *I_H2_* to the nearest *I_Sb_* site and then straight to the next nearest *I*_Sb_ site. Compared to the pristine monolayers, Li atoms display relatively higher diffusion barriers in the interlayer of G/Sb due to the larger binding energies, such as 0.59 eV and 0.80 eV for path-I and path-II, respectively. Such energy barriers (0.59–0.80 eV) are much lower than that in the interlayer space of bulk Sb (1.73 eV) [48] and are comparable to several commonly studied anode materials, such as high-capacity bulk silicon anode materials with a diffusion barrier around 0.57 eV, commercial graphite (0.5 eV) [49], and graphene/V_2_CO_2_(0.60 eV) [21], suggesting that antimonene covered with graphene are promising candidates for anode materials in battery applications. The reasons for the different energy barrier are ascribed to the presence of the geometric constraint in G/Sb, which affects the binding energy of Li intercalation, interlayer separation, and distance between Li and host. In other words, the Li atom should travel between graphene and antimonene at the interlayer region instead of moving on top of the surface of graphene or antimonene. However, it also has been reported that the diffusion behavior of Li in the interlayer of G/phosphorene (0.12 eV) [20] and C_3_N/phosphorene (0.24 eV) [41] are very similar to that on pristine ML phosphorene (0.09 eV) [6], which mainly arises from the diffusion of Li occurring inside the groove of phosphorene rather than the interlayer region.

Previous experimental studies confirmed that weakening the interaction of Li ions with the constituent layers can effectively reduce the diffusion barriers of Li atom [24,50]. This can be achieved by fabricating pillar structures to enlarge the interlayer distance of graphene and antimonene with the help of intercalated molecules or carbon nanotubes [50]. For instance, adjusting the interlayer distance can also improve the storage capacity by multilayer adsorption between the layers, such as Ti_3_C_2_O_2_ intercalated by quinone molecules displays fast kinetics and enhanced Li storage capacity [25]. In order to provide further support for this statement, we have performed additional calculations by enlarging the G–Sb interlayer distance from the equilibrium value of 3.53 Å to 4.20 Å manually. In this case, we find that the binding energies for Li located at *I*_Sb_ and *I*_C_ are 2.70 eV and 2.41 eV, which are smaller than those for G/Sb with equilibrium interlayer distance. The diffusion energy barrier for Li diffusion along the path-I reduces from 0.59 eV to about 0.34 eV. These calculated results support the energetic stability of Li embedded in the G/Sb interface, and we also find that the increasing of the G–Sb vertical distance could remarkably reduce the energy barrier of Li diffusion, due to the weakening of the interactions between adjacent sheets. Meanwhile, more Li atoms can be intercalated in the interlayer region and the theoretical specific capacity of G/Sb can be estimated to be 369.03 mAh·g^−1^ when interlayer distance is increased to 4.20 Å, which is very close to that of the commercial graphite anode (372 mAh g^−1^) [49].

## 4. Conclusions

In summary, we have systematically investigated the G/Sb heterostructure as an anode material for LIBs based on first-principles calculations. The calculated results reveal the G/Sb heterostructure not only exhibits high structural stability with formation energy of about –30.43 meV per atom, which is higher (the absolute value) than that of the stacked graphene bilayer (–24.95 meV per atom), but also possesses improved stiffness accompanied by greatly enhanced Young’s modulus (armchair: 145.20, zigzag: 144.36 N m^−1^) compared with free standing antimonene (armchair: 16.70, zigzag: 13.29 N m^−1^). It is noted that the existence of graphene could effectively reduce the bandgap of the G/Sb heterostructure to 0.03 eV, indicating the electrical conductivity of hybrid heterostructure is significantly enhanced compared to pristine ML antimonene, which is beneficial for G/Sb heterostructure–based LIBs. It is also found that the Li adatom prefers to intercalate into the interlayer region with larger adsorption energy (2.92 eV) instead of incorporating at the outside surface of graphene (2.02 eV) or antimonene (2.23 eV), which indicated that the interlayer region of the G/Sb heterostructure could provide a unique space for the accommodation of Li atoms due to the synergistic effect. Furthermore, Li adatoms maintain the high diffusion mobility of Li on the outside surfaces of antimonene or graphene of G/Sb heterostructure, while the diffusion barrier of Li in the interlayer region increases to 0.59 eV from 0.35 eV for pristine ML antimonene. It results from the limited interlayer distance and common modulation of individual monolayers on the Li bonding strength and can be reduced to 0.34 eV simply by increasing the interlayer distance. Importantly, the in-plane lattice constant of G/Sb heterostructure is slightly increased by around 4.5% along the *a/b*-axis direction in the case of full lithiation, which is significantly smaller than that of few layers of antimonene. The interlayer distance of G/Sb also increases relatively slightly (0.22 Å) during the full lithiation process, which is much less than that of a few layers of antimonene (1.14 Å) reported in experiments. Therefore, the capping of graphene can effectively minimize the volumetric expansion during charging process, which is desirable for anode materials of LIBs. The theoretical specific capacity of G/Sb heterostructure (369.03 mAh g^−1^) is also improved compared to pristine antimonene (about 208 mAh g^−1^) after full lithiation. The calculated open circuit voltage (OCV) is in the range of 0.89 V to 0.11 V, which is within the desired potential range (0.10–1.00 V) for anode materials. Overall, our findings show that G/Sb exhibits good mechanical properties, good electrical conductivity, very small volume expansion, relatively high capacity, and suitable open circuit voltage, which demonstrates that G/Sb holds promise to be a good candidate as an anode material in LIBs. Moreover, these studies could enhance the understanding of 2D antimonene-based heterostructures, which is important for the rational design of high-performance electrode materials for LIBs.

## Figures and Tables

**Figure 1 nanomaterials-09-01430-f001:**
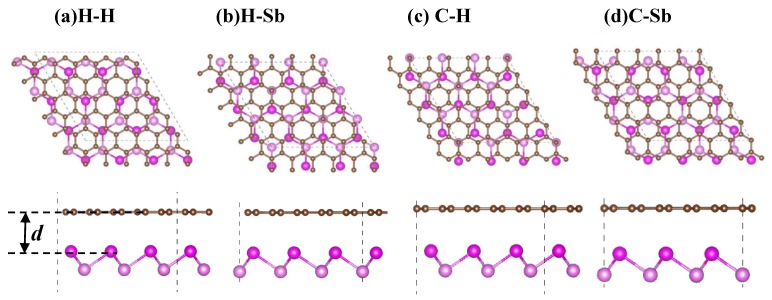
(**a**)–(**d**) Top and side views of four stacked patterns (H-H, H-Sb, C-H, and C-Sb) for graphene/antimonene heterostructures, in which the brown and pink balls represent C and Sb atoms, respectively. *d* represents the interlayer distance between graphene and antimonene. The dash lines in every stacked structure represent the rhombus primitive cell.

**Figure 2 nanomaterials-09-01430-f002:**
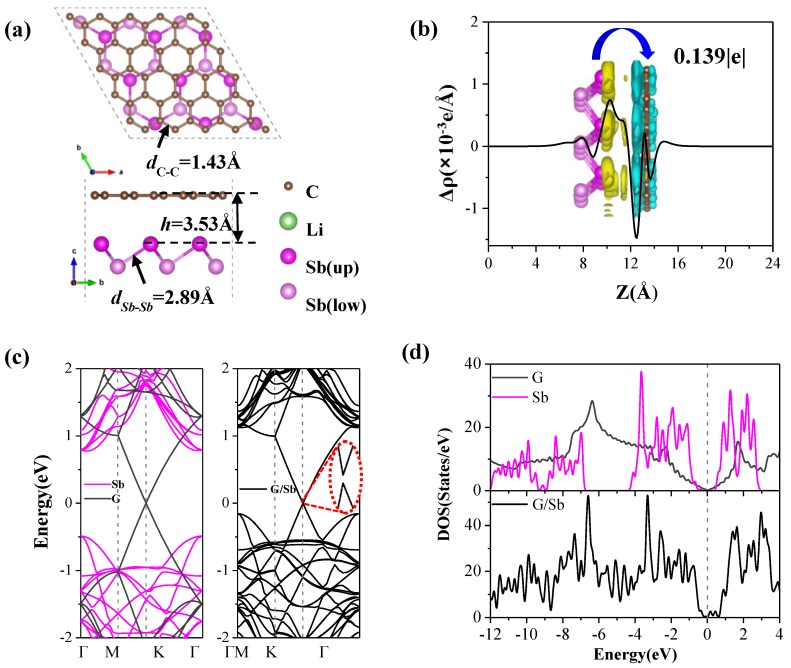
(**a**) Top and side views of the most stable configuration for G/Sb, (**b**) planar averaged charge density difference plots along z direction for G/Sb, and the corresponding charge density difference distribution (unit of isosurface level is 0.001eV/Å^3^), in which the yellow and green regions refer to the charge accumulation and depletion, respectively. (**c**) and (**d**)—band structure and density of states (DOS) for pristine monolayer of graphene and antimonene as well as the G/Sb heterostructure, respectively. The Fermi level is set to be 0 eV denoted by the black dash lines. The green balls represent Li atoms.

**Figure 3 nanomaterials-09-01430-f003:**
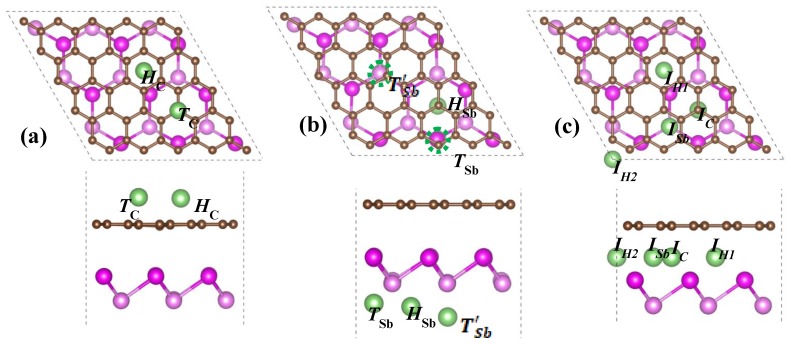
Top and side views of possible Li adsorption sites at the outside surface of (**a**) graphene and (**b**) antimonene, respectively, and (**c**) interlayer of G/Sb.

**Figure 4 nanomaterials-09-01430-f004:**
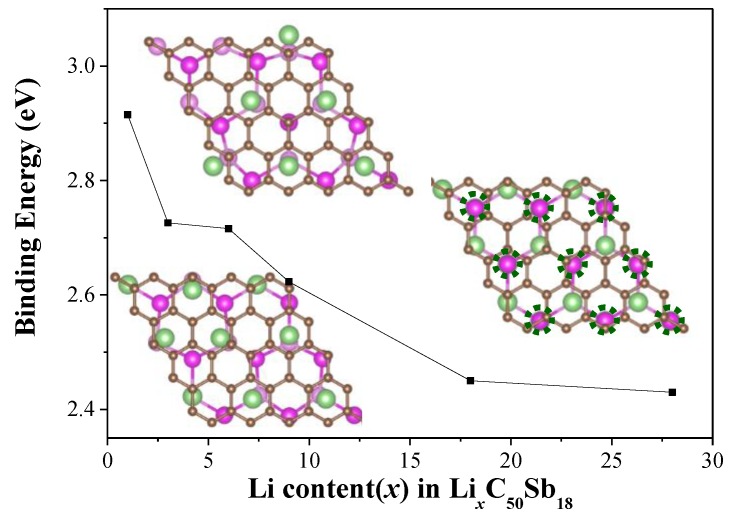
The binding energy of different number of Li atom adsorbed and intercalated on G/Sb (*x* = 1, 3, 6, 9, 18, and 28). The optimized stable configurations of Li*_x_*C_50_Sb_18_ (*x* = 6, 9 and 18) are shown.

**Figure 5 nanomaterials-09-01430-f005:**
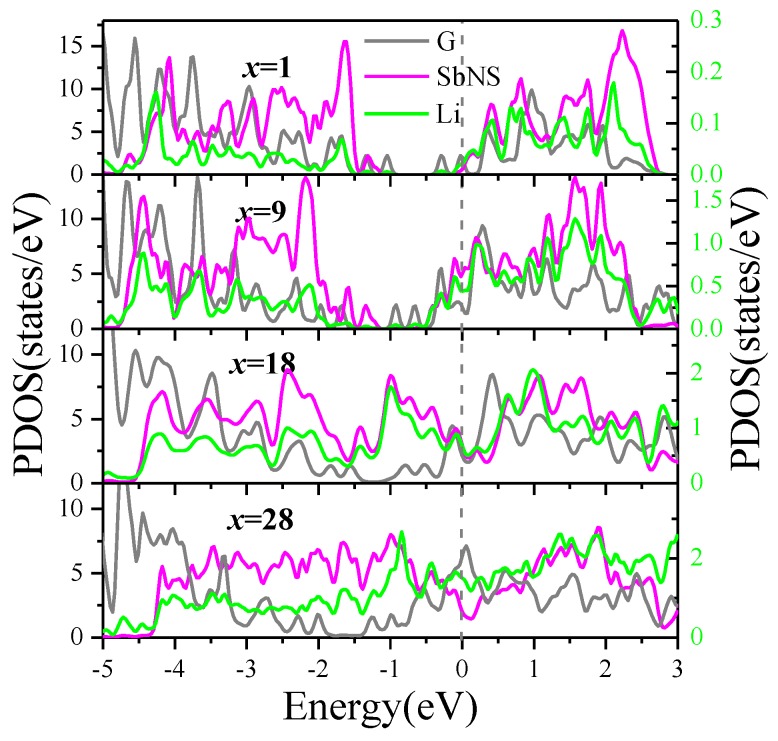
Density of states of Li*_x_*C_50_Sb_18_ (*x* = 1, 9, 18, 28). The scale of left Y-axis is for the PDOS of graphene and antimonene, which is represented by the gray and red lines, respectively; the scale of right Y-axis is for the PDOS of Li labeled by green line.

**Figure 6 nanomaterials-09-01430-f006:**
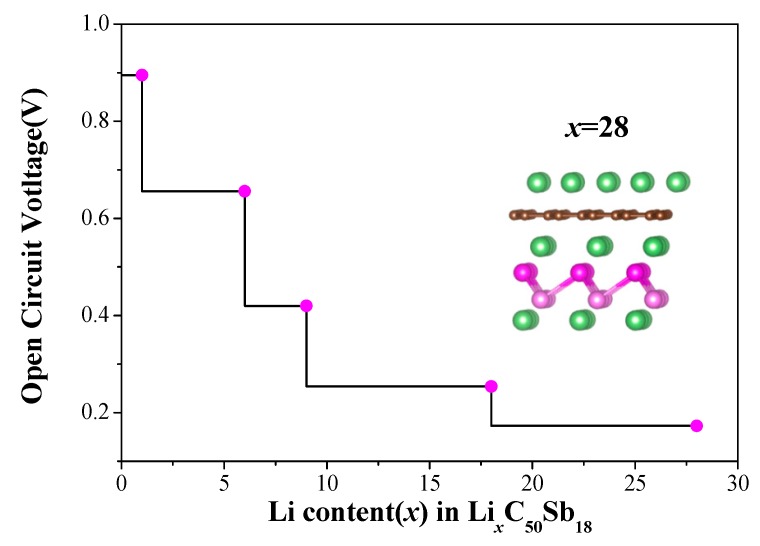
The calculated open circuit voltage (OCV) as a function of Li concentration in G/Sb. The insert is the stable configuration of G/Sb with 28 Li atoms intercalated.

**Figure 7 nanomaterials-09-01430-f007:**
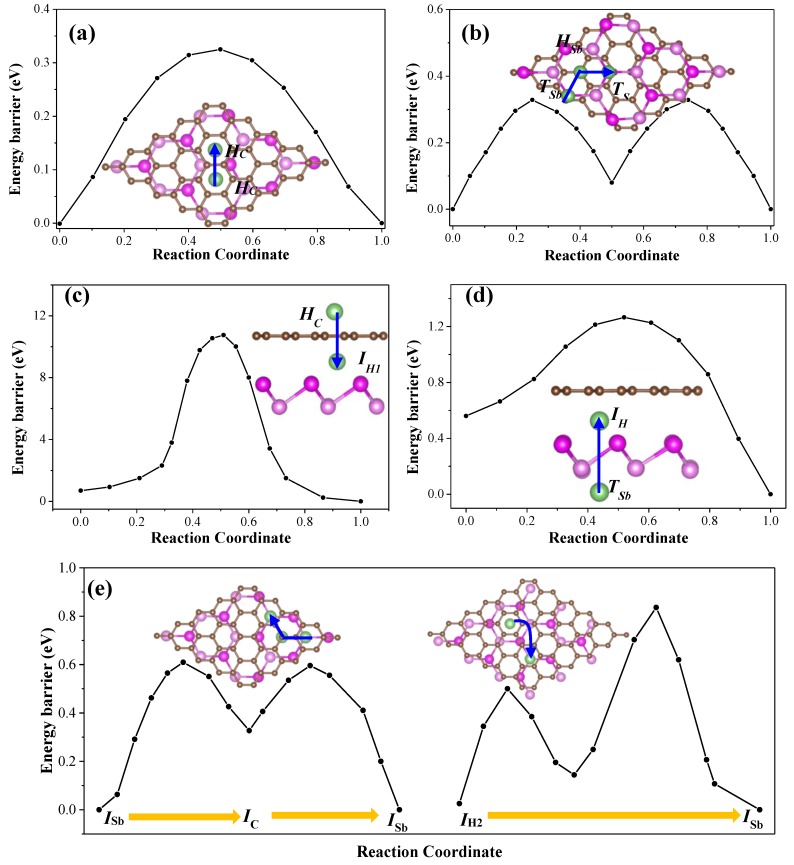
The possible diffusion paths and the energy profiles for Li diffusion in different regions of G/Sb: (**a**) outside surface of graphene and (**b**) antimonene; from outside surface of (**c**) graphene and (**d**) antimonene to the interlayer regions; (**e**) interlayer region between graphene and antimonone.

**Table 1 nanomaterials-09-01430-t001:** Calculated distance between graphene and antimonene (*d*) and the formation energy (*E_f_*) for different stacking vdW graphene/antimonene heterostructure and bilayer graphene and antimonene with an A–B stacking pattern. Distances and formation energy are given in units of Å and meV/atom, respectively.

	H-H	H-Sb	C-H	C-Sb	5 × 5-bi-G (AB)	3 × 3-bi-Sb (AB)
*D* (Å)	3.529	3.533	3.531	3.530	3.299	2.835
*E_f_* (meV/atom)	−30.43	−30.37	−30.35	−30.40	−24.95	65.33

**Table 2 nanomaterials-09-01430-t002:** Binding Energy *E_b_*(eV) of one Li adatom absorbed on monolayer graphene (Li/G) and antimonene(Li/Sb) and outside surface of graphene (Li/G/Sb) and antimonene (G/Sb/Li) of G/Sb heterostructure, intercalated in the interlayer site of G/Sb (G/Li/Sb), the calculated equilibrant interlayer distances ***d*_C-Sb_**(Å) between graphene and antimonene, and the averaged charge transfer for Li (ΔQ_Li_), carbon (ΔQ_C_), and antimony atoms (ΔQ_Sb_) for the above systems.

	Li Site	*E*_b_ (eV)	ΔQ_Li_	ΔQ_C_	ΔQ_Sb_	*d* _C-Sb_
Li/G	*H_C_*	1.90	+0.889	−0.889		
	*T_C_*	1.53	+0.901	−0.901		
Li/Sb	*T_Sb_*	1.96	+0.847	−0.847		
	*T’_Sb_*	1.24	+0.866	−0.866		
	*H_Sb_*	1.85	+0.855	−0.855		
Li/G/Sb	*H_C_*	2.02	+0.888	−1.043	+0.155	3.496
	*T_C_*	1.66	+0.902	−1.058	+0.156	3.497
G/Sb/Li	*T_Sb_*	2.23	+0.843	−0.518	−0.325	3.448
	*T’_Sb_*	1.42	+0.866	−0.516	−0.351	3.444
	*H_Sb_*	2.15	+0.853	−0.479	−0.374	3.453
G/Li/Sb	*I_H1_*	2.80	+0.846	−0.708	−0.138	3.535
	*I_H2_*	2.88	+0.844	−0.703	−0.141	3.515
	*I_Sb_*	2.92	+0.849	−0.797	−0.053	3.473
	*I_C_*	2.67	+0.842	−0.656	−0.186	3.520

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
