# Peer review of "Potential Application of Graphene/Antimonene Herterostructure as an Anode for Li-Ion Batteries: A First-Principles Study"

_nanomaterials, 2019, doi:10.3390/nano9101430_

Round 1
Reviewer 1 Report
Ping Wu et al studied the graphene/antimonene herterostructure for use as the anode in Li-ion batteries by first-principles calculation. It was reported that graphene plays the role for enhancing performance by exhibiting good mechanical properties, electrical conductivity and avoiding volume expansion. The study is interesting and useful for the rational design of anode materials based on hybrid 2D materials. The reviewer recommends for consideration of publishing the work, however a number of issues that could be addressed before decision.
The authors reported that the hybrid of graphene/antimonite herterostructure with few-layer materials, and compared with multilayer antimonene. How many layers for each type? What if the calculation for single layer of each, and what is different for few-layer and then multilayer? How is the organization of the graphene/antimonene herterostructure? What are they are randomly overlapping? And what if they are in order layer-by-layer? There are a number of typos and terminology such as electronical conductivity, electrode conductivityAuthor Response
Please see the attachment.

Reviewer 2 Report
Potential application of graphene/antimonene herterostructure as an anode for Li-ion batteries
The manuscript should be strongly revised, more discussions should be added. all below comments should be added.
The title should be changed and it should be cleared that it is not experimental. The abstract should be revised. More details of the results should be added. Page 2: line 91? open circle voltage??? The method section should be completed. A space should be between units and values. 4: right side Y-axis? Section 3.4 needs more discussions. How is the comparison with experimental results? The literature review should be improved.Author Response
Please see the attachment

Round 2
Reviewer 1 Report
The reviewer recommends for a publication of the work.
Author Response
In our revised manuscript, we have polished our the English language.
(1) We corrected the typo, eg. replace " Additionly " by "Additionallly" in line 23 of Abstract;(2) Change some expression, eg. change " compared with antimonene" to "compared with antimonene" line 18 of abstract section. (3) We also change the tense of the sentences the same as the context. eg. "were greatly enhanced " is changed to "are greatly enhanced".
Some sentences of the conclusion have been rewrote. See the text with underlines in conclusion for details.
Reviewer 2 Report
The revised manuscript was improved by the authors and they have responded to my comments.
Author Response
We double check the typo in our manuscript and tried our best to polish the English language in our revised manuscript. (1) We corrected the typo, e.g. replace " Additionly " by "Additionallly" in line 23 of Abstract; (2) Change some expression, e.g. change " compared with antimonene" to "compared with antimonene" line 18 of abstract section. (3) We also change the tense of the sentences the same as the context. e.g. "were greatly enhanced " is changed to "are greatly enhanced".